# The Importance of Suppressing Pathological Periostin Splicing Variants with Exon 17 in Both Stroma and Cancer

**DOI:** 10.3390/cells13171410

**Published:** 2024-08-23

**Authors:** Kana Shibata, Nobutaka Koibuchi, Fumihiro Sanada, Naruto Katsuragi, Yuko Kanemoto, Yasuo Tsunetoshi, Shoji Ikebe, Koichi Yamamoto, Ryuichi Morishita, Kenzo Shimazu, Yoshiaki Taniyama

**Affiliations:** 1Department of Advanced Molecular Therapy, Graduate School of Medicine/Faculty of Medicine, Osaka University, Suita 565-0871, Japan; shibata@amt.med.osaka-u.ac.jp (K.S.); koibuchi@cgt.med.osaka-u.ac.jp (N.K.);; 2Department of Clinical Gene Therapy, Graduate School of Medicine/Faculty of Medicine, Osaka University, Suita 565-0871, Japan; 3Department of Breast and Endocrine Surgery, Graduate School of Medicine/Faculty of Medicine, Osaka University, Suita 565-0871, Japan; 4Department of Geriatric and General Medicine, Graduate School of Medicine/Faculty of Medicine, Osaka University, Suita 565-0871, Japan; 5Graduate School of Dentistry (Oral and Maxillofacial Surgery), Osaka Dental University, Hirakata 573-1121, Japan

**Keywords:** periostin, breast cancer, stroma, cancer-associated fibroblasts (CAFs), tumor-associated macrophages (TAMs), tumor microenvironment (TME)

## Abstract

Background: Periostin (POSTN) is a type of matrix protein that functions by binding to other matrix proteins, cell surface receptors, or other molecules, such as cytokines and proteases. POSTN has four major splicing variants (PN1–4), which are primarily expressed in fibroblasts and cancer. We have reported that we should inhibit pathological POSTN (PN1–3), but not physiological POSTN (PN4). In particular, pathological POSTN with exon 17 is present in both stroma and cancer, but it is unclear whether the stroma or cancer pathological POSTN should be suppressed. Methods and Results: We transplanted 4T1 cells (breast cancer) secreting POSTN with exon 17 into 17KO mice lacking POSTN exon 17 to suppress stromal POSTN with exon 17. The results show that 17KO mice had smaller primary tumors and fewer metastases. Furthermore, to suppress cancer POSTN with exon 17, 4T1 cells transfected with POSTN exon 17 skipping oligo or control oligo were transplanted from the tail vein into the lungs. The results show that POSTN exon 17 skipping oligo significantly suppressed lung metastasis. Conclusions: These findings suggest that it is important to suppress POSTN exon 17 in both stroma and cancer. Antibody targeting POSTN exon 17 may be a therapeutic candidate for breast cancer.

## 1. Introduction

Breast cancer is the most commonly diagnosed cancer worldwide and the most common solid tumor in women. Its classification is based on immunohistochemical staining results for estrogen receptor (ER), progesterone receptor (PR), and human epidermal growth factor-2 (HER2) receptor expression. Specifically, breast cancers lacking expression of ER, PR, and HER2 are classified as a subtype called triple negative breast cancer (TNBC), which accounts for 15% to 20% of all breast cancers [1]. TNBC has a higher incidence in younger patient populations and may carry germline mutations in the tumor suppressor genes BRCA1 or BRCA2 [2]. Despite the continued introduction of innovative therapies to combat breast cancer, the basis of TNBC treatment still relies on conventional cytotoxic chemotherapy protocols using anthracyclines and taxanes [3]. The development of new therapeutic agents is needed.

On the other hand, periostin (POSTN) is a member of the matricellular proteins, whose functions are achieved by binding to matrix proteins as well as to cell surface receptors, or to other molecules, such as cytokines and proteases that interact, in turn, with the cell surface, with common property, high levels of expression during embryonic development, and in response to injury [4,5]. The expression of POSTN is associated with chronic inflammatory diseases, such as heart failure [6], diabetic retinopathy [7,8], asthma [9,10], allergic conjunctivitis, chronic sinusitis/chronic rhinosinusitis with nasal polyps, and atopic dermatitis [10], which are increasingly being observed. Furthermore, POSTN plays an important role in the pathogenesis of these diseases and significantly contributes to disease progression.

In addition, POSTN has four main splicing variants: PN1 (full-length *POSTN*), PN2 (*POSTN* lacking exon 17), PN3 (*POSTN* lacking exon 21), and PN4 (*POSTN* lacking exons 17 and 21) (Figure 1). We call PN1–3 the pathological POSTN, and PN4 the physiological POSTN. The role of POSTN in carcinogenesis is well known, and effective therapies targeting POSTN have been reported. Clinical data also report poor prognosis in 1086 TNBC patients with high POSTN expression [11]. POSTN plays important roles in various processes of tumorigenesis and metastasis, including the induction of epithelial–mesenchymal transition (EMT), enhancement of Akt and Erk activity in cancer stem cells, and involvement of the PI3K/Akt/survivin pathway in POSTN-mediated chemotherapy resistance [12,13,14]. 

However, the inhibition of all POSTN (PN1–4) has been reported to aggravate cancer in several animal models. Especially, PN4 is considered to be essential for suppressing tumor growth via making fiber capsule surrounding cancer cells, and should not be inhibited in cancer [15] Based on the above, we have reported the importance of inhibiting only pathological POSTN (PN1–3) but not physiological POSTN (PN4) [6,7,8,13,16,17,18], and especially, we showed the inhibition effect of the antibody for POSTN exon 17, which PN1 and PN3 have, on mice breast cancer cells [16]. Pathological POSTN was mainly secreted from stroma surrounding cancer cells [17] and some of the cancer cells themselves [16], but it is unclear whether the stroma or cancer pathological POSTN should be suppressed.

The purpose of this study is to elucidate the effect of pathological POSTN with exon 17 inhibition in stroma or cancer separately. 

## 2. Materials and Methods

### 2.1. Cell Culture

4T1 (CRL-2539, ATCC Manassas, VA, USA), 4T1/luc (CRL-2539-LUC2, ATCC Manassas, VA, USA), MDA-MB-231 (HTB-26, ATCC Manassas, VA, USA) mouse or human triple-negative breast cancer cell lines, or NIH3T3 (CRL-1658, Manassas, VA, USA) mouse fibroblast cell lines were cultured in Dulbecco’s modified Eagle’s medium (DMEM/Nacalai Tesque, Fushimi city, Kyoto, Japan) (26252-94, Nacalai Tesque, Fushimi city, Kyoto, Japan) supplemented with 10% fetal bovine serum (FBS) (26140079, Thermo Fisher Scientific, Grand Island, NY, USA)) and 1% penicillin–streptomycin (26140079, Thermo Fisher Scientific, Grand Island, NY, USA). NIH3T3 POSTN KO cells (3T3 Pn delete) were also established from NIH 3T3 cells using the CRISPR-cas9 method for all POSTN.

### 2.2. Quantitative Real-Time PCR

For reverse transcription, total RNA from cells or tissues was prepared as described previously. The RNA was quantified, and its integrity was confirmed. We used the SuperScript^TM^ III First-Strand Synthesis System (18080051, Thermo Fisher Scientific, Grand Island, NY, USA) with RNase Inhibitor for synthesizing cDNA, and the Applied Biosystems Viia 7 (Thermo Fisher Scientific, Grand Island, NY, USA) was used for detection following the manufacturer’s instructions. In each experiment, mouse glyceraldehyde-3-phosphate dehydrogenase (GAPDH) was amplified as a reference standard. 

GAPDH (5′-GAAGCAGGCATCTGAGGGC-3′, 5′-TTGAAGTCGCAGGAGACAACC-3′),

PN1 (5′-ATAA CCAAAGTCGTGGAACC-3′, 5′-TGTCTCCCTGAAGCAGTCTT-3′),

PN2 (5′-CCATGACTGTCTATAGACCTG-3′, 5′-TGTCTCCCTGAAGCAGTCTT3′),

PN3 (5′-ATAACCAAAGTCGTGGAACC-3′, 5′-TTTGCAGGTGTGTCTTTTTG-3′),

PN4 (5′-CCCCATGACTGTCTATAGACC-3′, 5′-TTCTTTGCAGGTGTGTCTTTT-3′),

IL6 (5′-TCCAGTTGCCTTCTTGGGAC-3′, 5′-TGTGAAGTCTCCTCTCCGGA-3′),

IL8 (5′ATTCCCGCGTTAGTCTGGTG-3′, 5′-GGAGCAGTCACCTGTGAACA-3′).

### 2.3. In Vivo Mouse Experiment

All animal care and studies were performed in accordance with protocols approved by Osaka University. The 4T1 cells metastasize from primary mammary tumors to multiple organs, including the lungs, and metastatic lesions closely resemble spontaneous human breast cancer arising from primary tumors [19]. These 4T1 cells (1 × 10^5^) or MDA-MB-231 cells suspended in PBS were injected into the mammary glands of BALB/c mice (7-week-old females), POSTN exon 17 KO mice or POSTN (1–4) KO mice [8], or NOD/Si-scid,IL-2RγKO mice, and tumor size was measured. The mice were then euthanized under appropriate anesthetic control. Tumor volume (mm^3^) was calculated as 1/2 × length (mm) × width (mm)^2^ [16]. Lung metastases were counted by staining with Bouin’s fixative and counting the number of lung colonies.

### 2.4. In Situ Hybridization

We evaluated the expression patterns of *POSTN* exon 17 on tissues using the 4T1 mice TNBC implanted syngeneic model. We stained *POSTN* exon 17 mRNA by in situ hybridization (ISH), using BaseScope^TM^ (Advanced Cell Diagnostics, Newark, CA, USA). This was performed according to the manual at Advantech Co., Ltd. (Tokyo, Japan). ISH was performed using the following probes. We designed a 1ZZZ probe called BA-Mm-Postn-1zz-st that targets 2162-2213 in NM_001368678.1 (POSTN exon 17 probe).

### 2.5. Organizational Assessment

Isolated perfused tumors were fixed in buffered 10% formalin, embedded in paraffin, and sliced into horizontal 5 μm sections for immunohistochemical and immunofluorescence staining. At least four fields of view per tumor were quantified using the ImageJ program downloaded from the website of the National Institutes of Health (NIH, Bethesda, MD, USA) to calculate the area positive for α-SMA, CD163, and vimentin in the tumor. Anti-smooth muscle actin antibody (M085101-2/Agilent, Santa Clara, CA, USA), Anti-CD163 antibody (ab182422/Abcam, Rumpington Cambridge, UK), and Anti-Vimentin antibody (ab92547/Abcam, Rumpington, Cambridge, UK) were considered.

### 2.6. Co-Culture Assay

4T1 mice TNBC cells and NIH3T3 mice fibroblast cells (5 × 10^5^ cells/well) were seeded in 6-well plates, and the medium was replaced with FBS-free medium after 24 h. Before the medium exchange, cells were centrifuged to ensure that they did not enter the medium. RNA was collected from each cell 24 h after each medium exchange and analyzed.

### 2.7. Skipping Oligo e Oligo

A morpholino antisense oligonucleotide that skips exon 17 of the mouse *POSTN* gene was purchased from Funakoshi. The nucleotide sequence is shown below. 

Mouse *POSTN* exon 17 skipping oligo: 5′-TGCTGAAAACATAGAAAGTGGAGCA-3′ Control oligo: 5′-CCTCTTACCTCAGTTACAATTTATA-3′ [20]. 

### 2.8. Anti-Human POSTN Exon 17 Antibody

In order to raise the mouse monoclonal antibody against exon 17 of human POSTN, the exon 17 peptide was synthesized at BIO MATRIX RESEARCH. The antibody was generated in immunized mice as previously described [7].

### 2.9. Statistical Analysis

The statistical analysis results are shown as the means ± SD. The Mann–Whitney test was performed for comparing multiple treatment groups. For the statistical analysis of the expression change of two groups, the Wilcoxon signed-rank test was performed.

### 2.10. Ethical Statement

All experimental procedures were approved by the Institutional Animal Committee of the Department of Veterinary Medicine, Faculty of Medicine, Osaka University, and followed the recommendations of the *Guidelines for Animal Experiments in Research Institutions* (MEXT, Osaka), *Guidelines for Animal Experiments in Research Institutions* (MHLW), and *Guidelines for the Proper Conduct of Animal Experiments* (Science Council of Japan). The following were used in this study: wild-type BALB/c mice were purchased from Charles River Corporation, and POSTN exon 17 KO mice and POSTN (1–4) KO mice were generated at RIKEN [8].

## 3. Results

### 3.1. Analysis of POSTN Alternative Splicing Variants in 4T1 Mice TNBC

We performed real-time PCR using 4T1 mice TNBC (Figure 2A). The PN3 splicing variants containing exon 17 showed significantly higher expression than the other splicing variants. Conversely, PN2 and PN4, which lack exon 17, showed lower expression levels than PN3. Next, we performed ISH using the tissue from 4T1 implanted mice samples (Figure 2B). Subsequent ISH showed that pathological POSTN exon 17 was expressed not only in the cancer cells but also in the stroma surrounding cancer. On the right side of the yellow dotted line, spindle-shaped fibroblast-dominated stromal tissue lacking nuclear atypia is depicted. On the left side of the yellow dotted line, on the other hand, all exon 17 positive cells are cancer cells, and isolated, relatively small nuclear, red-stained cells infiltrate the stroma. We calculated the percentage of exon 17-positive cells in cancer cells (5.6% ± 4.4), and stroma (2.5% ± 0.8).

In addition, exon 17 positive cells have invasive capacity in cancer. Taken together, the contrasting features observed between stromal tissue and exon 17 positive cancer cells highlight the complex interactions between cancer cells and their surrounding microenvironment. These findings underscore the importance of studying such cell–cell interactions to better understand the mechanisms that drive tumor progression and invasion. After verifying *POSTN* splicing variant expression, we planted to evaluate the effect of the inhibition of pathological POSTN with exon 17 in stroma and cancer separately.

### 3.2. Evaluation for the Pathological POSTN in Exon 17 KO Mice

To evaluate the inhibition effect of POSTN exon 17 in stroma, we use POSTN exon 17 KO mice (17KO) [8]. Tumor microenvironment (TME) refer to cancer stroma, reactive stroma, and cancer-associated fibroblasts (CAFs), which exist in close proximity to the cancer epithelium. Both stromal and epithelial cell phenotypes are known to be fibroblasts that comprise the coevolving cancer stroma during tumorigenesis and produce a variety of growth factors that promote cancer cell growth [21]. Next, we measured CAFs and TAMs in the TME using 4T1 implanted models of wild-type and 17KO mice (Figure 3). 17KO mice had predominantly suppressed α-SMA positive cells and vimentin positive cells (CAFs) compared to wild-type mice (Figure 3A,B). In addition, 17KO mice had predominantly suppressed CD163 positive cells, a marker of tumor-associated macrophages (TAMs) compared to wild-type mice (Figure 3C). TAMs are known to interact with cancer cells to promote tumor invasion [22]. Taken together, the inhibition of *POSTN* exon 17 in stroma significantly decreased CAFs and TAMs, improving the TME. Next, we calculated primary tumor size and metastasis (Figure 3D–F). In addition, to replicate a previous report suggesting that PN4 should not be suppressed for cancer growth [15], we compered the results between POSTN (1–4) KO mice and 17KO mice. As a result, although POSTN (1–4) KO mice significantly increased primary cancer growth (*p* < 0.05) and suppressed lung metastasis (*p* < 0.05) in Figure 3F, 17KO mice significantly suppressed primary cancer growth (*p* < 0.05) and lung metastasis (*p* < 0.05) compared to wild-type mice (Figure 3D,E). These results suggest that the suppression of *POSTN* exon 17 but not *POSTN* (1–4) in stroma is beneficial in the size of murine TNBC primary tumor, while the suppression of both of them are beneficial in lung metastasis. On the other hand, tumors were increased in POSTN (1–4) KO mice compared to wild-type mice. On the contrary, lung metastasis was significantly suppressed in POSTN (1–4) KO mice compared to wild-type mice (Figure 3F). 

### 3.3. Evaluation for the Pathological POSTN with Exon 17 Inhibition in Cancer

To evaluate the inhibition effect of *POSTN* exon 17 in cancer, we emoloyed exon skipping into cancer cells. At first, we compared *POSTN* (1–4) variants expression among no-treatment NIH 3T3 mice fibroblasts (positive control), *POSTN* (1–4) all knock out 3T3 cells by crisper cas9 (negative control), control oligo transfected 3T3 cells (control), and *POSTN* exon 17 skipping antisense transfected 3T3 cells. It can be confirmed that PN 1 and 3 has been erased by exon 17 skipping. (Figure 4A) Next, we transfected control oligo or *POSTN* exon 17 skipping oligo into 4T1-Luc cells. In order to metastasize to the lung, we administered these cells from the tail vein. The inclusion of *POSTN* exon 17 skipping oligo resulted in a marked inhibition of lung metastasis. This result suggests that the suppression of *POSTN* exon 17 in cancer is also beneficial in murine TNBC (Figure 4B).

### 3.4. The Role of Pathological POSTN with Exon 17 Inhibition in Fibroblasts and Cancers In Vitro

To investigate the role of the POSTN with exon 17 inhibition in stroma fibroblasts and cancer in vitro, additional experiments were performed with NIH3T3 mouse fibroblasts and 4T1 mice TNBC (Figure 5A–D). When we added 4T1 mice TNBC cells’ supernatant into NH3T3 cells, IL-6 and IL-8 were significantly increased, and the POSTN exon 17 antibody treatment reduced them significantly (**; *p* < 0.05 vs. without POSTN exon 17 antibody) (Figure 5A,B). Similarly, when we added NH3T3 cells’ supernatant into 4T1 mice TNBC cells, *IL-6* and *IL-8* were significantly increased, and the POSTN exon 17 antibody treatment reduced them significantly (**; *p* < 0.05 vs. without POSTN exon 17 antibody) (Figure 5C,D). These result suggest that POSTN with exon 17 is secreted in fibroblasts and cancer, and both of them significantly increase IL-6 and IL-8 in each other’s cells.

### 3.5. The Role of Pathological POSTN with Exon 17 Inhibition in Stroma and Cancer In Vivo

Finally, to investigate the role of the POSTN with exon 17 inhibition in both stroma and cancer in vivo, we treated POSTN exon 17 antibody (40, 100, 250, and 600 μg/mice) or control IgG (600 µg/mice) with the MDA-MB 231 human TNBC xenograft model. Previously, we reported that PN17-Ab detected pathological POSTN with exon 17 in MDA-MB 231 breast cancer cells by western blotting [18]. We had confirmed the expression of pathological POSTN with exon 17 in MDA-MB231 cells. POSTN exon 17 antibodies significantly inhibited primary tumor growth in a dose-dependent manner (*p* < 0.05) (Figure 6A,B), as well as lung metastasis (*p* < 0.05) (Figure 6C,D). The inhibition of POSTN exon 17 in both stroma and cancer by antibody is beneficial in murine TNBC.

## 4. Discussion

POSTN was first identified in osteoblast cell lines and showed effects on bone regeneration. The N-terminal region of *POSTN* (exons 1–15), including the EMI domain and the four FAS1 domains, is conserved in various species, while the C-terminal region (exons 16–23), primarily PN1–4, undergoes alternative splicing. PN1 represents full-length *POSTN*, PN2 represents *POSTN* lacking exon 17, PN3 represents *POSTN* lacking exon 21, and PN4 represents *POSTN* lacking exons 17 and 21 (Figure 1). It was reported that the C-terminal region of *POSTN* is likely to be a key to disambiguating POSTN function [23].

We call PN4 the physiological POSTN, and PN1–3 the pathological POSTN. The POSTN region around exons 17 and 21 undergoes extensive alternative splicing [24]. We have reported that PN1–4 expression had a relationship with jawbone growth by micro-computed tomography analysis [25].

In addition, POSTN is a member of the matricellular proteins, whose functions are achieved by binding to matrix proteins as well as to cell surface receptors, or to other molecules, such as cytokines and proteases that interact, in turn, with the cell surface with common property, high levels of expression during embryonic development, and in response to injury [4,5]. The expression of POSTN is associated with chronic inflammatory diseases, such as heart failure [6,7,26,27], diabetic retinopathy [8], cancer [13,16,17,18], stroke [28,29], osteoarthritis (OA) [30], asthma [9,10], allergic conjunctivitis, chronic sinusitis/chronic rhinosinusitis with nasal polyps [31], and atopic dermatitis [10], which are increasingly being observed. Furthermore, POSTN plays an important role in the pathogenesis of these diseases and significantly contributes to disease progression.

The role of POSTN in carcinogenesis is well known, and effective therapies targeting POSTN have been reported. Clinical data also report poor prognosis in 1086 TNBC patients with high POSTN expression [11]. POSTN plays important roles in various processes of tumorigenesis and metastasis, including induction of epithelial–mesenchymal transition (EMT), enhancement of Akt and Erk activity in cancer stem cells, and involvement of the PI3K/Akt/survivin pathway in POSTN-mediated chemotherapy resistance [12,13,14]. POSTN is required for the maintenance of cancer stem cells and its function can be inhibited to prevent metastasis [32]. There are also reports of decreased TNBCs secreting POSTN, accompanied by decreased numbers of M2 tumor-associated macrophages and tumor blood vessels [17].

However, the inhibition of all POSTN (1–4) has been reported to aggravate cancer in several animal models, such as S180 sarcoma cells, B16F10 malignant melanoma cells, and LLC lung cancer. PN4 is considered to be essential for suppressing tumor growth via making fiber capsule surrounding cancer cells and should not be inhibited in cancer primary tumor [15]. Based on the above, we have reported the importance of inhibiting only pathological POSTN but not physiological POSTN (PN4) [6,7,8,13,16,17,18] and, especially, we showed the effect of the antibody for POSTN exon 17 on the inhibition of mice 4T1 breast cancer cells proliferation, migration, invasion and bone destruction [16]. Pathological POSTN was secreted from CAFs in stroma surrounding cancer cells and cancer cells themselves [11,21], but it is unclear whether the stroma or cancer pathological POSTN should be suppressed.

In this study, we evaluated the importance of inhibiting stromal or cancer pathological POSTN with exon 17 separately. We produced 17KO mice lacking pathological POSTN with exon 17, and POSTN(1–4) KO mice lacking PN1–4 in the whole body [8]. We used pathological POSTN with exon 17-expressed 4T1 mice TNBC [19], where metastases arise spontaneously from primary tumors and progress to lymph nodes and other organs, mirroring human breast cancer [19]. We proved that the pathological POSTN with exon 17 inhibition in stroma but not cancer significantly suppressed primary tumor growth and metastasis. In addition, we produced *POSTN* exon 17 skipping oligo with the exon skipping method to inhibit it in cancer cells [20]. We transfected this oligo to 4T1 mice TNBC cells and proved that the pathological *POSTN* with exon 17 inhibition in cancer but not stroma noticeably suppressed metastasis. Taken together, this proves for the first time the importance of inhibiting pathological POSTN with exon 17 in both stromal and cancer for the mice TNBC model separately. The antibody inhibiting both of them could be fit for this situation, and we proved that the antibody for pathological POSTN with exon 17 significantly suppressed the primary growth and metastasis in the MDA-MB 231 human TNBC xenograft model in a dose-dependent manner. In addition, we reproduced the past reports that suppressing all PN(1–4) increased cancer growth in primary tumor but decreased in lung metastasis number. Because PN4 is believed to make fiber capsule surrounding cancer cells, it may be reasonable that PN4 inhibition increases primary tumor growth but not lung metastasis number. The details will be left to future research.

Next, we evaluated the synergic effect between fibroblasts in stroma and cancer. The results (Figure 3) show that the pathological POSTN with exon 17 from both fibroblasts and cancer significantly increased IL-6 and IL-8 in each other’s cells. IL-6 and IL-8 are important cytokines that increase cancer cell activity in many cancers and are involved in tumorigenesis and metastasis [21]. CAFs are central players in the TME of solid tumors, and pathological POSTN with exon 17 regulating them [21,33].

Previous reports have highlighted POSTN’s role in activating the ERK signaling pathway and regulating the transcription of key cytokines, like IL6 and IL8 via NF-κB, which influences the downstream activation of STAT3 [22]. Similar mechanisms may operate involving POSTN exon 17. These results confirm that, when cells come into contact with cancer cells or fibroblast culture supernatants, the cancer microenvironment is formed, and inflammatory cytokines are elevated. Furthermore, the antibody for POSTN exon 17 suppressed inflammation in TME (Figure 7).

Recently, Trundle et al. reported that the pathological *POSTN* with exon 17 inhibition by skipping oligo improved in the D2.mdx. mice model of Duchenne muscular dystrophy (DMD). The grip strength of the *POSTN* exon 17 skipping oligo transfected mice was stronger than that in the wild-type level. In addition, they performed bioinformatic analysis of *POSTN* variants, resulting in the change of binding site of TGF-b1 and POSTN. POSTN exon 17 has a significant impact on the structure of the POSTN protein [34]. According to our result, the effects of both *POSTN* exon 17 skipping and the antibody for POSTN exon 17 are similar, but it will be clarified in the future how it is appropriate to suppress POSTN exon 17 for each disease.

Moreover, Rusbierg-Weberskov et al. performed the POSTN structural analysis and showed that the C-terminal domain lacked a tertiary structure and combined 143 proteins directly [35]. The C-terminal of POSTN may have more multi-functions than we can imagine. In the future, the C-terminal potential of periostin will be increasingly elucidated.

## 5. Patents

The patent of the Ex17 antibody belongs to Osaka University and Periotherapia Co., which has the priority negotiation right.

## Figures and Tables

**Figure 1 cells-13-01410-f001:**
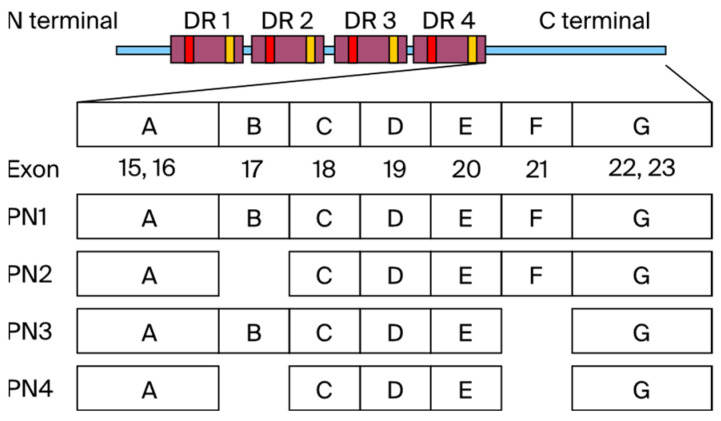
The N-terminus of POSTN has EMI domain and 4 repeat domains (FAS1). The C-terminal region (exons 15–23) centered on PN1–4 undergoes alternative splicing. Pathological POSTN splicing variants include exons 17 and 21 (PN1–3), while physiological POSTN lacks POSTN exon 17 and 21 (PN4).

**Figure 2 cells-13-01410-f002:**
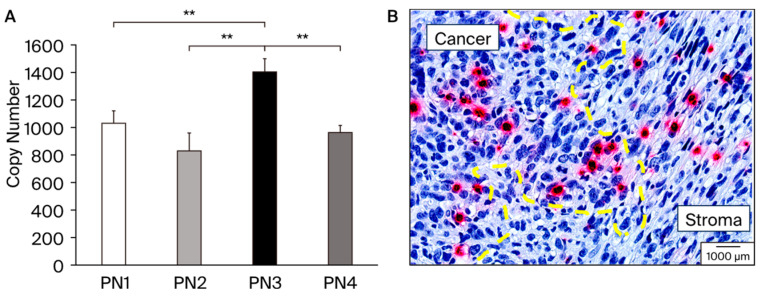
(**A**): Expression patterns of various *POSTN* splicing variants (PN1–4), which were analyzed in 4T1; a cultured cell line of TNBC model RNA was isolated from the 4T1 cell line and absolute quantification of each *POSTN* splicing variants was performed using PCR method. PN3 was significantly expressed from 4T1, **; *p* < 0.05 vs. PN1, 2, and 4. Results are shown as absolute values and expressed as mean ± standard error. (**B**): A sample from a study performed 21 days after transplantation of 4T1 cells into BALB/c mice. The primary tumor was excised and fixed with 4% paraformaldehyde. Yellow dot line separates cancer and stroma area. The pathology specialists from external agencies, Applied Medical Research Laboratory (Osaka, Japan) distinguished as cancer and stroma, and drew the yellow dot line. Breast cancer cells with large round shapes; strong dysplastic nuclei and alveolar formation can be distinguished. Red color shows the *POSTN* exon 17 expression in cancer cells and fibroblast cells in stroma. The black scale bar in the figure represents 1000 μm and serves as a size reference.

**Figure 3 cells-13-01410-f003:**
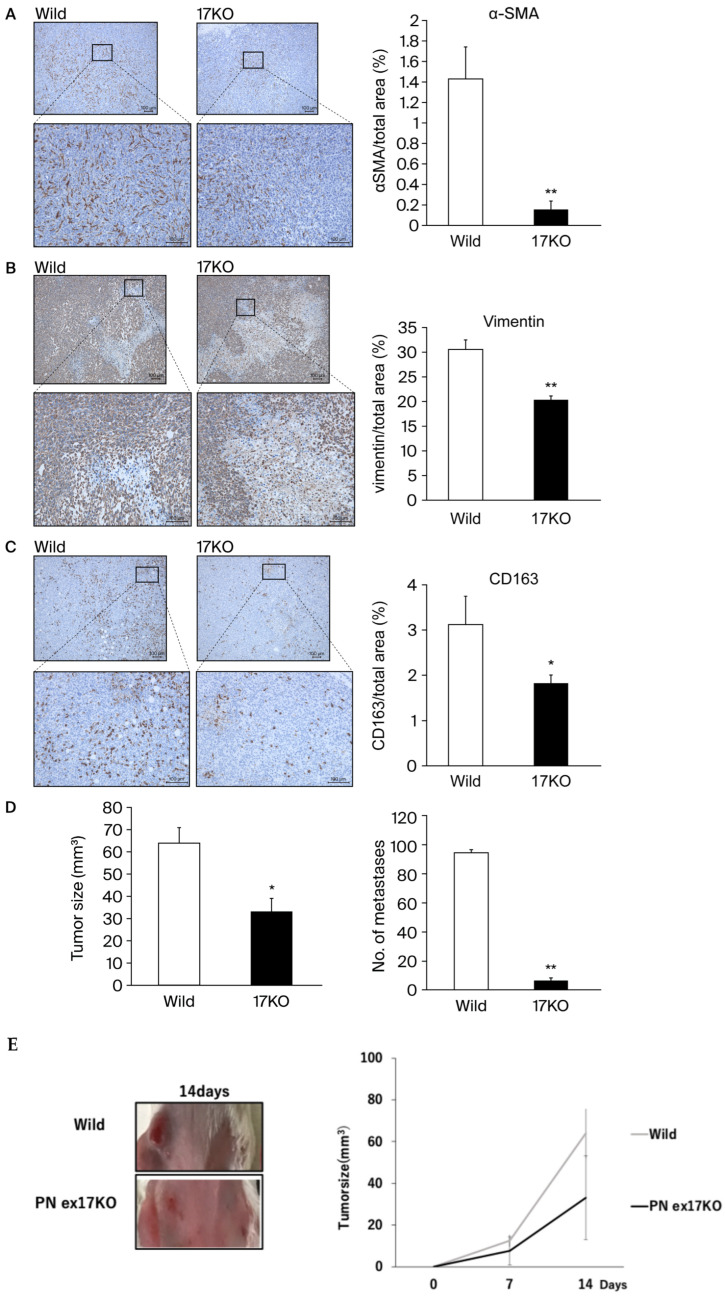
(**A**–**D**): The 1 10^5^ 4T1 cells suspended in PBS were injected into the mammary glands of wild-type mice (7-week-old female BALB/c) and POSTN exon 17KO mice (7-week-old female). (**A**–**C**): 17KO mice had significantly suppressed a-SMA positive cells or vimentin positive cells (CAFs), and CD163 positive cells (TAMs) compared to wild-type mice, **; *p* < 0.01 vs. wild-type mice. (**D**–**E**): Tumors were predominantly suppressed in 17KO mice compared to wild-type mice. Lung metastasis was also significantly suppressed in 17KO mice compared to wild-type mice. (**F**): Tumors were increased in POSTN (1–4) KO mice compared to wild-type mice. On the contrary, lung metastasis was significantly suppressed in POSTN (1–4) KO mice compared to wild-type mice. Lung metastasis was evaluated by staining with Bouin fixation and counting the number of lung colonies. Tumor size was measured on days 14, 21, and 28, with results shown for day 28. Relative values are shown as mean ± standard error (*n* = 4~7, ** *p* < 0.01 or * *p* < 0.05 vs. wild mice). **; *p* < 0.01 vs. wild-type mice. Primary tumors in mice were evaluated by calculating tumor volume (mm^3^) as 1/2 × length (mm) × width (mm). Results are shown as absolute values and expressed as mean ± standard error. In the representative images, the yellow scale bar is shown at 100 μm.

**Figure 4 cells-13-01410-f004:**
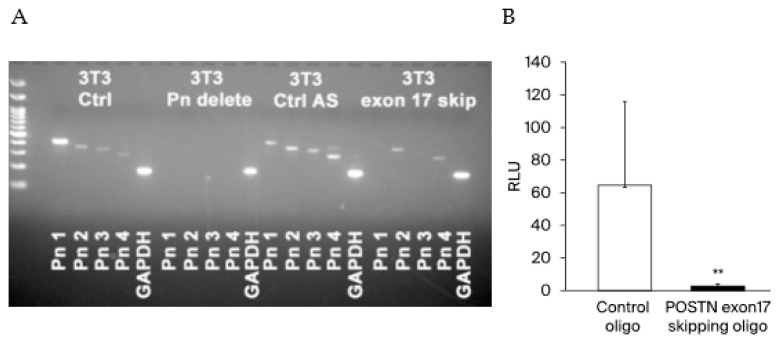
(**A**) Using 3T3 fibroblasts, we checked PN 1–4. PN1 and 3 were prominently decreased after exon 17 skipping. compared to 3T3 Ctrl and 3T3 Ctrl antisense. It can be confirmed that PN 1 and 3 have been erased by exon 17 skipping. 3T3 Ctrl: no-treatment 3T3 fibroblasts, 3T3 Pn delete: *POSTN* (1–4) all knock out 3T3 cells by CRISPR-Cas9, 3T3 Ctrl AS: control antisense transfected 3T3 cells, 3T3 exon 17 skip: exon 17 skipping antisense transfected 3T3 cells. (**B**) We transfected control oligo or *POSTN* exon 17 skipping oligo to 4T1-Luc cells. In mice treated with control oligo-transfected 4T1-Luc cells, high luciferase activity was measured, indicating that 4T1-Luc cells metastasize to the lungs. On the other hand, in mice treated with *POSTN* exon 17 skipping oligo transfected 4T1-Luc cells, luciferase activity was very low, indicating that 4T1-Luc cells rarely metastasize to the lungs. These results suggest that skipping oligo that selectively skip *POSTN* exon 17 inhibit metastasis of breast cancer cells to the lung (each *n* = 3~6, **; *p* < 0.01 vs. control). For comparison between the two groups, the Mann–Whitney test (MWU) was used.

**Figure 5 cells-13-01410-f005:**
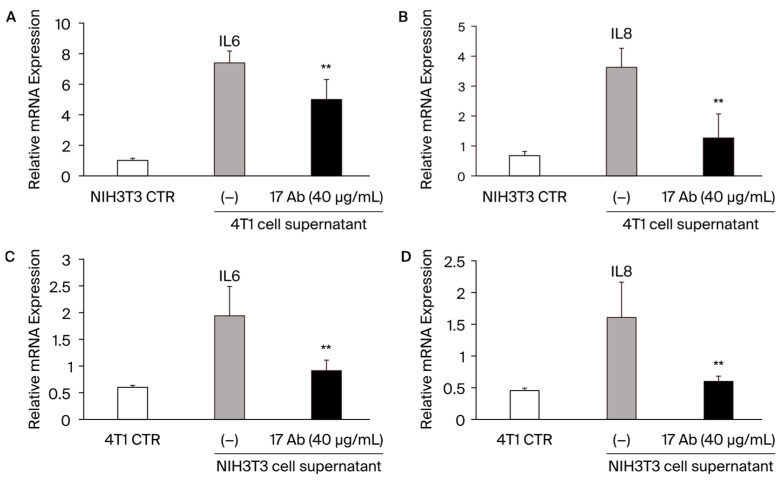
(**A**–**D**): When we add 4T1 mice TNBC cells supernatant into NH3T3 cells, *IL-6* and *IL-8* are significantly increased, and the POSTN exon 17 antibody treatment reduces them significantly. **; *p* < 0.05 vs. POSTN exon 17 antibody non-treatment. Results are shown as absolute values and expressed as mean ± standard error.

**Figure 6 cells-13-01410-f006:**
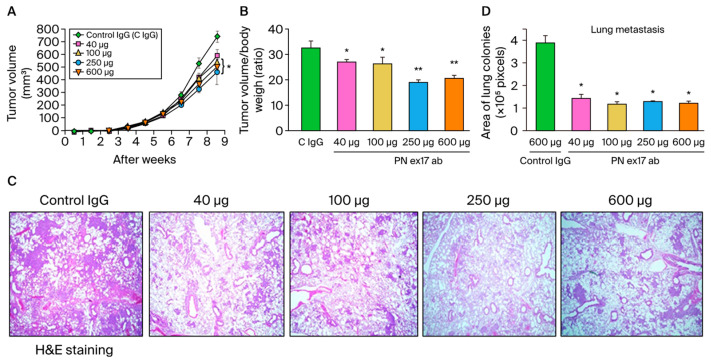
(**A**–**D**) We show the result of PN17-Ab on xenograft model. A total of 1 × 10^6^ MDA-MB-231 cells were suspended in PBS and injected into the chest of 7-week-old female NOD/Si-scid,IL-2RγKO mice. Once the tumor volume reached 100 mm^3^, the POSTN exon 17 antibody (PN17-Ab, 40–600 μg/mice) or mouse IgG antibody (Control IgG, 600 μg/mice) was administered once weekly. (**A**,**B**): After dissection up to 9 weeks after transplantation, the POSTN exon 17 antibody suppressed primary tumor growth in a dose-dependent manner (*n* = 6, *; *p* < 0.05, **; *p* < 0.01 vs. control IgG). Tumor size (mm^3^) was calculated as 1/2 × width (mm) × length (mm). We evaluated lung metastasis after H and E staining. (**C**,**D**): Area of lung colonies were significantly reduced by the treatment of the POSTN exon 17 antibody (*n* = 6, *; *p* < 0.05, **; *p* < 0.01 vs. control IgG). Three site-specific analyses were performed from one lung section. Results are shown as absolute values and expressed as mean ± standard error.

**Figure 7 cells-13-01410-f007:**
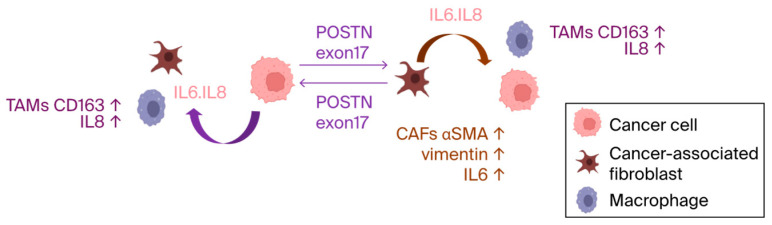
The pathological POSTN with exon 17 was secreted from both fibroblasts in stroma and cancer. Once secreted, it increased IL-6 and IL-8, which induces the inflammation of the TME with TAMs and may promote malignancy of cancer.

## Data Availability

The data that support the findings of this study are available from the corresponding author, Y. Taniyama, upon reasonable request.

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
