# Peer review of "The Importance of Suppressing Pathological Periostin Splicing Variants with Exon 17 in Both Stroma and Cancer"

_cells, 2024, doi:10.3390/cells13171410_

Round 1

Reviewer 1 Report

Comments and Suggestions for Authors

In the research article titled “The Importance of Suppressing Pathological Periostin Splicing Variants with Exon 17 in Both Stroma and Cancer” authors tried to delineate the pathological effects of Periostin (POSTN) splice variants in breast cancer. The study focused mainly on the inhibition of POSTN in cancer and stroma compartments separately to see which of these contributes to the pathology. The authors found that inhibiting POSTN in both cancer and stroma are essential to control tumor growth. The article is fairly well written with a methodology that justifies their findings. I recommend this article for publication with few minor corrections and experimental validations given below which would strengthen the findings.

1.      In Figure 2B, authors could include a proliferation marker like Ki67 to demarcate cancer cells from the surrounding stroma. At the moment it is unclear how the cells are distinguished as cancer and stroma given that both the cell types express POSTN. 

2.      For figure 3D. authors should consider including the tumor size of POSTN ex17 KO mice vs wildtype mice along all the data points collected. It will be valuable to see if the temporal effect of POSTN on tumor growth. Also, would be valuable to include the expression of all four POSTN variants to show that only pathological POSTN is responsible for the tumor growth inhibition.   

3.      In figure 4, It would be beneficial to see the expression of individual POSTN splicing variant when authors used the skipping oligo to inhibit the expression of pathological POSTN.  

4.      In figure 6, when authors performed inhibition of POSTN expression in both fibroblast and cancer cells, it would be interesting to see if the tumor growth reduction happens by cellular apoptosis or if it happens by immune cell mediated killing. Staining for immune compartment would add value.  

Author Response

Reviewer 1

We appreciate your comments and criticisms of our manuscript. We are very happy to hear that “ The article is fairly well written with a methodology that justifies their findings.”. We have improved our manuscript in compliance with your suggestions.

Comment 1:

In Figure 2B, authors could include a proliferation marker like Ki67 to demarcate cancer cells from the surrounding stroma. At the moment it is unclear how the cells are distinguished as cancer and stroma given that both the cell types express POSTN. 

Response 1:

As you indicated, it was difficult to distinguished as cancer and stroma in detail , so we asked Mr. Takayoshi Seko and Mr. Kotaro Isoda from Applied Medical Research Laboratory, the pathology specialists from external agencies, to distinguished them. They mentioned that breast cancer cells with large round shapes, strong dysplastic nuclei and alveolar formation can be distinguished. According to your suggestion, we add following sentences.

“The pathology specialists from external agencies, Applied Medical Research Laboratory (Osaka, Japan) distinguished as cancer and stroma, and drew the yellow line. Breast cancer cells with large round shapes, strong dysplastic nuclei and alveolar formation can be distinguished. ” in Fig 2 diagram

In addition, the reviewer 1 recommended IHC of Ki67 to distinguished as cancer and stroma, but fibroblasts in stroma was also proliferating. Indeed, we need additional over 1 month to add Ki67 IHC, but we must return in 10 days.

Comment 2:

For figure 3D. authors should consider including the tumor size of POSTN ex17 KO mice vs wildtype mice along all the data points collected. It will be valuable to see if the temporal effect of POSTN on tumor growth. Also, would be valuable to include the expression of all four POSTN variants to show that only pathological POSTN is responsible for the tumor growth inhibition.   

Response 2:

According to reviewer’s suggestion, we changed Fig 3D to show temporal effect for primary tumor. We analyzed POSTN (1-4) all variants in 4T1 breast cancer in vitro but did not measure them in mice after 4T1 transplantation. We previously reported an increase in PN1 over time after 4T1 implantation (INTERNATIONAL JOURNAL OF MolecularMedicine28: 181-186, 2011), so believe POSTN (1-4) all variants expression in mice.

Comment 3:

In figure 4, It would be beneficial to see the expression of individual POSTN splicing variant when authors used the skipping oligo to inhibit the expression of pathological POSTN. 

Response 3:

According to reviewer’s suggestion, we added following sentences in 3.3. Evaluation for the Pathological POSTN with Exon 17 Inhibition in Cancer

At first, we compared POSTN (1-4) all variants expression among no-treatment 3T3 mice fibroblasts (positive control), POSTN all knock out 3T3 by crisper cas9 (negative control), control oligo transfected 3T3 (control), and POSTN exon 17 skipping oligo transfected 3T3. It can be confirmed that PN 1 and 3 has been erased by exon 17 skipping.

We added following sentence in Fig 4 (A)

Using 3T3 fibroblasts, we checked PN 1-4. PN1 and 3 after exon 17 skipping were prominently decreased compared to those of 3T3 Ctrl and 3T3 Ctrl oligo. It can be confirmed that PN 1 and 3 has been erased by exon 17 skipping.

3T3 Ctrl: no-treatment 3T3 fibroblasts, 3T3 Pn delete: POSTN knock out 3T3 by crisper cas9, 3T3 Ctrl AS: control oligo transfected 3T3, 3T3 exon 17 skip: exon 17 skipping oligo transfected 3T3

Comment 4:

In figure 6, when authors performed inhibition of POSTN expression in both fibroblast and cancer cells, it would be interesting to see if the tumor growth reduction happens by cellular apoptosis or if it happens by immune cell mediated killing. Staining for immune compartment would add value.

Response 4:

Thank you for your advice, but we cannot perform them in 10 days. The purpose of this study is to clarify whether periostin should be suppressed, cancer or stroma. We would like to clarify the mechanism in future research.

Reviewer 2 Report

Comments and Suggestions for Authors

The manuscript demonstrated that it is important to inhibit POSTN exon 17 in both stroma and cancer and targeting POSTN exon 17 may be a therapeutic candidate for breast cancer. The work is interesting for the research field of POSTN in cancer. However, there are some concerns should be addressed.

1. As the physiological POSTN, the expression cell types, location and physiological function of PN4 should be elucidated  in the introduction section. 

2. The staining of Figure 2B can’t distinguish cancer cells and stromal cells enough well, so whether exon 17-positive cells are cancer cells or stromal cells is difficult to be recognized. Applying the corresponding epithelial cell marker and stromal cell marker staining in the serial tissue sections may be a fine method.

3. From Figure 2B, exon 17-positive cells by ISH in cancer cells or stromal cells are both a small number of cells. The percentage of exon 17-positive cells in cancer cells or stromal cells should be assessed.

4. In 4T1 implanted wild-type and 17KO mice models (Figure 3D), the authors only showed the statistical analysis results of tumor size and metastasis. The organs with tumor nodules from primary tumor or metastatic sites or their sections should be also shown.

5. To evaluate the inhibition effect of POSTN exon 17 in cancer, the authors transfected control oligo or POSTN exon 17 skipping oligo into 4T1-Luc cells and administered these cells from the tail vein of mice. In order to exclude effect of POSTN exon 17 in stroma, this experiment should administer 4T1-Luc cells transfected control oligo or POSTN exon 17-skipping oligo into 17KO mice but no wild-type mice.

6. Similarly, the organs with metastatic tumor nodules or their sections should be shown in Figure 4.

7. MDA-MB-231 human TNBC xenograft model was performed. Before the experiment, the expression of PN3 in MDA-MB 231 cells should be detected.

8. The effect of POSTN exon 17 antibody on the cell proliferation, migration, invasion and/or other biological behaviors of breast cancer cells should be investigated.

Author Response

Reviewer 2 

We appreciate your comments and criticisms of our manuscript. We are very happy to hear that “ The work is interesting for the research field of POSTN in cancer.We have improved our manuscript in compliance with your suggestions.

Comment 1:

As the physiological POSTN, the expression cell types, location and physiological function of PN4 should be elucidated  in the introduction section. 

Response 1:

According to reviewer’s suggestion, we added following sentences in Introduction

However, the inhibition of all POSTN (1–4) has been reported to aggravate cancer in several animal models. Especially, PN4 is considered to be essential for suppressing tumor growth via making fiber capsule surrounding cancer cells and should not be inhibited in cancer [15] Based on the above, we have reported the importance of inhibiting only pathological POSTN but not physiological POSTN (PN4).

We also added the results of primary tumor size and lung metastasis between POSTN all KO mice and POSTN exon 17 KO mice after in Fig 3D.

We added following sentences in Results.

In addition, to reproduce the past report suggesting that PN4 should not be suppressed for cancer [15], we compared the results between POSTN (1-4) KO mice and 17KO mice. As a result, although POSTN (1-4) KO mice significantly increased primary cancer growth (p < 0.05) and suppressed lung metastasis (p < 0.05), 17KO mice significantly suppressed primary cancer growth (p < 0.05) and lung metastasis (p < 0.05) compared to wild-type mice. These results suggest that the suppression of POSTN exon 17 but not POSTN (1-4) in stroma is beneficial murine TNBC primary tumor size, while the suppression of both of them are beneficial in lung metastasis.

We added following sentences in Discussion.

In addition, we reproduced the past reports that suppressing all POSTN (1-4 ) variants causes cancer to growth in primary tumor, but not in lung metastasis. Because PN4 is believed to make fiber capsule surrounding cancer cells, it may be reasonable that PN4 inhibition increases primary tumor growth but not lung metastasis number. The details will be left to future research.

Comment 2:

The staining of Figure 2B can’t distinguish cancer cells and stromal cells enough well, so whether exon 17-positive cells are cancer cells or stromal cells is difficult to be recognized. Applying the corresponding epithelial cell marker and stromal cell marker staining in the serial tissue sections may be a fine method.

Response 2:

As you indicated, it was difficult to distinguished as cancer and stroma in detail , so we asked Mr. Takayoshi Seko and Mr. Kotaro Isoda from Applied Medical Research Laboratory, the pathology specialists from external agencies, to distinguished them. They mentioned that breast cancer cells with large round shapes, strong dysplastic nuclei and alveolar formation can be distinguished. According to your suggestion, we add following sentences.

“The pathology specialists from external agencies, Applied Medical Research Laboratory (Osaka, Japan) distinguished as cancer and stroma, and drew the yellow line. Breast cancer cells with large round shapes, strong dysplastic nuclei and alveolar formation can be distinguished. ” in Fig 2 diagram

Comment 3:

From Figure 2B, exon 17-positive cells by ISH in cancer cells or stromal cells are both a small number of cells. The percentage of exon 17-positive cells in cancer cells or stromal cells should be assessed.

Response 3:

We added the following sentences in 3. Results 3.1. Analysis of POSTN Alternative Splicing Variants in 4T1 Mice TNBC

We calculated the percentage of exon 17-positive cells in cancer cells (5.6%±4.4), and stroma (2.5%±0.8).

Comment 4:

In 4T1 implanted wild-type and 17KO mice models (Figure 3D), the authors only showed the statistical analysis results of tumor size and metastasis. The organs with tumor nodules from primary tumor or metastatic sites or their sections should be also shown.

Response 4:

We added the organs with tumor nodules from primary tumor in Fig 3E.

Comment 5:

To evaluate the inhibition effect of POSTN exon 17 in cancer, the authors transfected control oligo or POSTN exon 17 skipping oligo into 4T1-Luc cells and administered these cells from the tail vein of mice. In order to exclude effect of POSTN exon 17 in stroma, this experiment should administer 4T1-Luc cells transfected control oligo or POSTN exon 17-skipping oligo into 17KO mice but no wild-type mice.

Response 5:

Thank you for your advice, it is interesting to administer 4T1-Luc cells transfected control oligo or POSTN exon 17-skipping oligo into 17KO mice, but the purpose of this study is to clarify whether periostin should be suppressed, cancer or stroma. It will spend over 6 months to perform this experiment because we should make 17 KO mice from an embryo.

We would like to clarify it in future research,

Comment 6:

Similarly, the organs with metastatic tumor nodules or their sections should be shown in Figure 4.

Response 6:

According to reviewer’s suggestion, we added the organs with tumor nodules from primary tumor in Fig 3E, but don’t have the organs with metastatic tumor nodules. It spends over 6 months to perform this experiment.

Comment 7:

MDA-MB-231 human TNBC xenograft model was performed. Before the experiment, the expression of PN3 in MDA-MB 231 cells should be detected.

Response 7:

According to reviewer’s suggestion, we added following sentences in 3.5. The Role of Pathological POSTN with Exon 17 Inhibition in Stroma and Cancer In Vivo

Previously, we reported that PN17-Ab detected pathological POSTN with exon 17 in MDA-MB 231 breast cancer cells by western blotting. [18] We had confirmed the expression of pathological POSTN with exon 17 in MDA-MB231 cells.

Comment 8:

The effect of POSTN exon 17 antibody on the cell proliferation, migration, invasion and/or other biological behaviors of breast cancer cells should be investigated.

Response 8:

According to reviewer’s suggestion, we added following sentences in Discussion.

and especially, we showed the effect of the antibody for POSTN exon 17 on the inhibition of mice 4T1 breast cancer cells proliferation, migration, invasion and bone destruction [16].

Round 2

Reviewer 2 Report

Comments and Suggestions for Authors

The authors have addressed my concerns.